# OpenReview forum: "Vaccine: Perturbation-aware Alignment for Large Language Models against Harmful Fine-tuning Attack"
_NeurIPS.cc/2024/Conference — NeurIPS 2024 poster_

### Official Review · Reviewer_M5RT · 2024-06-13

**Soundness:** 3
**Presentation:** 3
**Contribution:** 3
**Rating:** 6
**Confidence:** 3

**Summary:**

This paper introduces Vaccine, a technique that improves the security of Large Language Models (LLMs) by incorporating perturbation-aware alignment during fine-tuning. Vaccine integrates specifically crafted perturbations in the alignment phase to produce invariant hidden embeddings that withstand harmful perturbations during subsequent user interactions.

**Strengths:**

The authors demonstrate with empirical evidence that even minimal harmful data can induce significant shifts in LLM embeddings, disrupting their alignment.

The method Vaccine was validated on widely-used LLMs, including Llama2, Opt, and Vicuna, showing substantial enhancements in their resilience against malicious prompts while preserving their reasoning capabilities with non-malicious prompts.

A work that may be interesting to the broad community.

**Weaknesses:**

The computational overhead of the method seems to linearly scales with model size.

When the harmful ratio is high, the method doesn't seem to perform very well.

**Questions:**

If the data for fine-tuning comes from the user, why would the user upload harmful data to break the model.

**Limitations:**

The authors have partially addressed the limitations of their work.

---

> ### Author Rebuttal · Authors · 2024-07-31
>
> We thank the reviewer for the generally positive review. As below, we show extra results and discussion to address the concerns.
>
> **W1: Computational overhead scales with model size**.
>
> **Step time**. Because Vaccine requires double forward-backward pass in each optimization step, the training time of Vaccine is approximately double compared to the SFT baseline. Because the training time itself is scaling with the model size, the overhead is also scaling with the model size.  See the following table for our evaluation results.
>
> |  | OPT-1.3b | OPT-2.7b | OPT-6.7b | OPT-13b |
> |---|---|---|---|:---:|
> | Running time(SFT) | 0.06 s (1x) | 0.08s (1x) | 0.09s (1x) | 0.12s (1x) |
> | Running time(Vaccine) | 0.11s (1.83x) | 0.15s (1.875x) | 0.17s (1.88x) | 0.24s (2x) |
> | Extra time overhead | 0.05s (0.83x) | 0.07s (0.875x) | 0.08s (0.88x) | 0.12s (1x) |
>
> However, it is important to note that **this is only a one-time cost** for aligning a pre-trained model. In the fine-tuning-as-a-service scenario, the same aligned model is used to finetune for all the finetuning requests (which arrive at the rate of thousands or millions/per hour). Since Vaccine does not incur extra overhead for fine-tuning, its computation overhead should not be a big concern for real-time service for a service provider.
>
> *Accelerated Vaccine*. It is also possible to accelerate Vaccine. Vaccine requires double training time because in every step we need to first search for the perturbation, and then apply the perturbation to the model to do another forward-backward pass. To address the reviewer's concern, we propose **accelerated Vaccine. The idea is simple- we search for the perturbation **not for every step**, but to do it for **every $\tau$ step**. See Algorithm 2 in our attached pdf for details.     The following table shows the evaluation results.
>
> | Methods | Harmful score | Finetune accuracy | Step time |
> |:---:|:---:|:---:|:---:|
> | SFT | 59.20 | 94.40 | 0.12902s (1x) |
> | Vaccine | 50.40 (+0) | 92.40 | 0.24852s (1.93x) |
> | Accelerated Vaccine($\tau=100$) | 51.00 (+0.60) | 95.20 | 0.13140s (1.02x) |
> | Accelerated Vaccine($\tau=1000$) | 52.00 (+1.60) | 94.20 | 0.12956s (1.004x) |
> | Accelerated Vaccine($\tau=10000$) | 53.20 (+2.80) | 94.80 | 0.12934s (1.002x)|
> | Accelerated Vaccine($\tau=20000$) | 58.80 (+8.40) | 94.40 | 0.12902s (1x) |
>
> Our results surprisingly show that by searching perturbation for every 100 steps, Acceleated Vaccine is still able to achieve decent defense (harmful score is increased by a marginal 0.60), but the step time is significantly shortened. We hope accelerated Vaccine can erase the reviewer's concern about resource overhead.
>
> **GPU memory**.  On the other hand, the extra memory overhead is scaled with the hidden embedding size, because in the second forward-backward pass we need to register the perturbation to the hidden embedding. The following table shows how the GPU memory usage scales with the hidden embedding size:
>
> |  | OPT-1.3B | OPT-2.7B | OPT-6.7B | OPT-13B |
> |---|---|---|---|---|
> | Hidden Embedding size | 24*2048 (1x) | 32*2560 (1.67x) | 32*4096 (2.67x) | 40*5120 (4.16x) |
> | Memory (SFT) | 5.586GB | 10.814GB | 25.469GB | 48.824GB |
> | Memory (Vaccine) | 5.596GB | 10.830 GB | 25.492GB | 48.863 GB |
> | Extra Memory Cost | 0.0097GB(1x) | 0.0157GB(1.62x) | 0.0235GB(2.42x) | 0.039GB(4.02x) |
>
> As shown, for OPT-13B, only a marginal 0.039/48.824=0.08% extra memory is induced compared to SFT. The memory cost is apparently marginal.
>
>
>
>
> **W2: Performance downgrades when the harmful ratio is high**.
>
> Indeed, Vaccine experiences performance downgrades when the harmful ratio is high. This however is an inevitable drawback of an alignent stage solution. Particularly, an alignment stage solution aims at enhancing the model's robustness to the harmful data, but fine-tuning on too much harmful data (e.g., pure harmful data) can still break the enhanced robustness.
>
> While we in this paper only aim to increase the aligned model's robustness in the alignment stage, we admit that this may not be enough for the hard case (e.g., a large harmful ratio). We envision that future research can build on top of Vaccine to provide a stronger defense.  We will discuss this in our limitation and future direction section.
>
> **Q1: Why would users upload harmful data to break the model?**
>
> There are two possible cases for the user to upload harmful data.
>
> i) **Non-adversary case**. The users are not aware that the data they upload contains harmful instances. Users may collect fine-tuned data from their application use case, and they may not carefully inspect and clean the data.
>
> ii) **Adversary case**. The user (e.g., a business adversary) aims to scandalize the service provider to provide harmful content (or political sensitivity content) from the API. Because the finetuned model is deployed on the service provider's server and the harmful content is transmitted from their API, the service provider may face legal accusations or governance issues. An example is that users may ask "How do you comment the war between Israel and Palestine"? The service provider is responsible for the answer.
>
> While the first case may be more realistic and common, the second case may raise serious concerns for the service provider.

---

> > ### Comment · Reviewer_M5RT · 2024-08-08
> >
> > Thanks to the authors for clarifying my confusion. I am keeping my score unchanged for now.

---

> > > ### Author Response · Authors · 2024-08-08
> > > **Thanks for the feedback!**
> > >
> > > We thank the reviewer for acknowledging our efforts in rebuttal! Please feel free to leave us a comment if you need charification during our interaction with other reviewers.

---

### Official Review · Reviewer_CLDA · 2024-07-04

**Soundness:** 3
**Presentation:** 3
**Contribution:** 3
**Rating:** 6
**Confidence:** 3

**Summary:**

This paper proposes a novel alignment technique called "Vaccine", which addresses the security risks of large language models (LLMs) during user fine-tuning. It is found that even a small amount of harmful data can destroy the alignment effect of a model, leading to the "alignment destruction effect".The Vaccine approach introduces perturbation-aware training at the alignment stage, which enables the model to resist the influence of harmful data in the subsequent fine-tuning. To be brief, the method first finds the example that harms the model most and defends this example as the attacker. Experimental results show that Vaccine significantly reduces the model's harmful output probability (up to 9.8%) while maintaining good downstream task performance. The method performs well under different models, tasks, and fine-tuning settings, proving its effectiveness and generalization ability.

**Strengths:**

The strength of this paper lies in the following two main points:
1. the authors first present a sample attack scenario under "Finetuning as a service", and discuss the need to maintain security under this scenario.
2. the authors propose the phenomenon of "embedding drift" as the main cause of broken alignments qualitatively.
3. the simplicity and generality of the proposed approach. Specifically, the method proposed in the paper combines the idea of adversarial training, and requires only one more gradient finding and parameter optimization, while applying to various alignment methods. This method can be used as a plug-and-play method in different application scenarios.

**Weaknesses:**

The weakness of this paper may be the following:
1. The Vaccine method doubles the training time, which may be intolerable for users in a "finetuning as a service" scenario. Related acceleration methods can be considered. I am willing to raise my score once this is solved.
2. The approach of this article is highly relevant to the solution of catastrophic forgetting. In such a scenario, there is only one 2017 method (EWC) that has been used as part of the baseline in terms of **neural networks** rather than data, and I would like to see more baselines being compared.
3. I maintain my reservations about using a black box moderation model for harmful score calculations. Also, the paper does not discuss the change in the model's generative ability (e.g., ppl) after using the vaccine method.

**Questions:**

I am very interested in the phenomenon of embedding drift and noticed that you have demonstrated it using the L2 norm. can you provide a more intuitive visualisation such as t-SNE?

**Limitations:**

The authors have adequately addressed the limitations in the checklist guidelines.

---

> ### Author Rebuttal · Authors · 2024-08-04
>
> We sincerely thank the reviewer for the constructive review comments. As belows, we try to address the separate comment.
>
> **W1: Double training time**
>
> Indeed, Vaccine requires double training time, because in every step we need to first search for the perturbation, and then apply the perturbation to the model to do another forward-backward pass. To address the reviewer's concern, we propose **accelerated Vaccine. The idea is simple- we search for the perturbation **not for every step**, but to do it for **every $\tau$ steps**. See Algorithm 2 in our attach pdf for details.     The following table for the evaluation results.
>
> | Methods | Harmful score | Finetune accuracy | Step time |
> |:---:|:---:|:---:|:---:|
> | SFT | 59.20 | 94.40 | 0.12902s (1x) |
> | Vaccine | 50.40 (+0) | 92.40 | 0.24852s (1.93x) |
> | Accelerated Vaccine($\tau=100$) | 51.00 (+0.60) | 95.20 | 0.13140s (1.02x) |
> | Accelerated Vaccine($\tau=1000$) | 52.00 (+1.60) | 94.20 | 0.12956s (1.004x) |
> | Accelerated Vaccine($\tau=10000$) | 53.20 (+2.80) | 94.80 | 0.12934s (1.002x)|
> | Accelerated Vaccine($\tau=20000$) | 58.80 (+8.40) | 94.40 | 0.12902s (1x) |
>
> Our results surprisingly show that by searching perturbation for every 100 steps, Acceleated Vaccine is still able to achieve decent defense (harmful score is increased by a marginal 0.60), but the step time is significantly shortened.
>
>
>
> **W2: Lack of baseline**
>
> Indeed, not enough baselines are used in this submission, partly because this work is one of the initial works for the harmful fine-tuning issue. After NeurIPS submission deadline, we do see there are a few defense solutions arise, e.g., RepNoise[1]].  RepNoise, same as Vaccine, is an alignment stage solution that aims at improving the neural network's robustness, which makes it a perfect baseline to compare.   In the following table, we show the performance comparison under different harmful ratios.
>
> |  | Harmful Score | --> | --> | --> | --> | Finetune Accuracy | --> | --> | --> | --> |
> |---|:---:|:---:|:---:|:---:|:---:|:---:|:---:|:---:|:---:|:---:|
> |  | p=0 | p=0.01 | P=0.05 | p=0.1 | p=0.2 | p=0 | p=0.01 | P=0.05 | p=0.1 | p=0.2 |
> | SFT | 54.20  | 52.40  | 54.80  | 58.80  | 64.80  | 93.20  | 94.40  | 94.20  | 94.40  | 94.20  |
> | Repnoise | 51.20  | 50.20  | 51.20  | 52.40  | 62.40  | 92.80  | 92.80  | 92.60  | 92.40  | 92.80  |
> | Vaccine | 45.40  | 46.40  | 48.40  | 50.40  | 60.00  | 92.00  | 91.80  | 92.00  | 92.40  | 93.40  |
>
> As shown, Vaccine consistently outperforms RepNoise with a smaller harmful score and also a higher finetune accuracy.
>
> **W3-A: Evaluation using black box moderation model**
>
> We totally agree that using a moderation model is not ideal and not accurate enough for performance evaluation. However, at the current stage, there is not an accurate benchmarking method to evaluate the model's harmfulness. Existing research, e.g., [2][3] utilizes GPT4 for evaluation. RepNoise [1] follows our setting to use BeaverTails's moderation model for evaluation. We generally believe that compared to GPT, the moderation model we use is less "black box", given that GPT4 is not even open-sourced, and the score it gives may change due to version updates or randomness.
>
>
> **W3-B: the paper does not discuss the change in the model's generative ability (e.g., ppl)**
>
> For model evaluation, we mainly use finetune accuracy to evaluate the model's reasoning ability (for example, for GSM8k, we measure the accuracy based on whether the model can predict the final answer). Indeed, it is interesting to study the model's generative ability using metrics like perplexity. In the following table, we show the comparison results after the model fine-tuned on wikitext.
>
> |  | Harmful score |  -->| --> |  -->|  -->| Perplexity (lower the better) | --> |-->  | --> |  -->|
> |---|:---:|:---:|:---:|:---:|:---:|:---:|:---:|:---:|:---:|:---:|
> |  | p=0 | p=0.01 | P=0.05 | p=0.1 | p=0.2 | p=0 | p=0.01 | P=0.05 | p=0.1 | p=0.2 |
> | SFT | 54.60  | 55.60  | 58.00  | 59.00  | 63.40  | 26.08  | 26.11  | 26.29  | 26.61  | 27.27  |
> | Repnoise | 54.20  | 53.80  | 55.80  | 57.60  | 59.20  | 35.32  | 36.11  | 37.16  | 36.64  | 38.68  |
> | Vaccine | 46.60  | 46.20  | 48.40  | 50.80  | 55.40  | 34.36  | 34.45  | 34.67  | 34.85  | 35.77  |
>
> The results show that Vaccine may slightly increase the perplexity of the model compared to SFT. However,  in comparison, RepNoise also increases the perplexity but is less effective in maintaining a low harmful score.
>
> **Q1: Intuitive visualization of harmful embedding drift**
>
> We thank the reviewer for the suggestion of t-SNE visualization. We plot the embedding drift of SFT and Vaccine under different harmful ratios. When the harmful ratio is high, it is intuitive to see that the embedding is drifting toward a specific direction. Interestingly, the embedding drift of Vaccine is slighter, making the drifted embedding still able to preserve the alignment knowledge. This may better explain how Vaccine really works.
>
> [1] Rosati D, Wehner J, Williams K, et al. Representation noising effectively prevents harmful fine-tuning on LLMs[J]. arXiv preprint arXiv:2405.14577, 2024.
>
> [2] Hsu C Y, Tsai Y L, Lin C H, et al. Safe LoRA: the Silver Lining of Reducing Safety Risks when Fine-tuning Large Language Models[J]. arXiv preprint arXiv:2405.16833, 2024.
>
> [3] Qi X, Zeng Y, Xie T, et al. Fine-tuning aligned language models compromises safety, even when users do not intend to![J]. arXiv preprint arXiv:2310.03693, 2023.

---

> ### Comment · Reviewer_CLDA · 2024-08-08
>
> Thank you for your detailed and constructive rebuttal, which includes extensive experimental evidence.
>
> I am particularly impressed by your introduction of the **accelerated** Vaccine method. The results demonstrating only a 2% overhead when $\tau = 100$ with minimal performance degradation significantly reduce the application cost of the Vaccine method. I highly recommend including this improvement in the manuscript for future versions.
>
> Based on my initial review and your comprehensive response, I decided to adjust my score positively.
>
> P.S. There appears to be a typo in Algorithm 2 of the attached PDF: "t mod $\gamma$ == 0" should be reviewed for accuracy.
>
> Best regards,

---

> > ### Author Response · Authors · 2024-08-08
> > **Thanks for the re-evaluation of our work**
> >
> > We thank the reviewer for the encouraging comment, and also for positively adjusting the socre reflecting our efforts in rebuttal!
> >
> > It is a a great relief for us to see that accelerated Vaccine address your main concern!  We will definitely include the accelerated Vaccine as well as its evaluation into the next revision. Indeed, there is a typo in Algorithm 2 of the attached PDF. It should be $t$ mod $\tau$==0. We will fix it in our revision. Thank you for pointing this out!
> >
> > Please feel free to leave us a comment if you need charification during our interaction with other reviewers.

---

### Official Review · Reviewer_Pifs · 2024-07-14

**Soundness:** 3
**Presentation:** 3
**Contribution:** 3
**Rating:** 6
**Confidence:** 3

**Summary:**

This paper presents a novel approach to enhance the security of finetuning-as-a-service for Large Language Models (LLMs). The proposed method, Vaccine, introduces a perturbation-aware alignment technique to mitigate the risk of harmful data introduced during user finetuning. The paper demonstrates that Vaccine can effectively maintain alignment robustness against harmful prompts while preserving reasoning abilities on benign prompts.

**Strengths:**

Quality: The paper provides comprehensive empirical evidence demonstrating the effectiveness of Vaccine in reducing harmful scores while maintaining finetuning accuracy across multiple models and datasets.
Clarity: The methodology is well-explained, with clear descriptions of the problem, the proposed solution, and the experimental setup.

**Weaknesses:**

Resource Overhead: The increased computational and memory overhead introduced by Vaccine, though justified, might be a limitation for practical deployment, especially in resource-constrained environments. For example the runtime and memory usage should be compared to normal finetune and other baselines.

**Questions:**

Could you provide more information about Resource Overhead?

**Limitations:**

Yes

---

> ### Author Rebuttal · Authors · 2024-08-01
>
> We thank the reviewer for the positive comments on our work. Below we try to address the concern on **resource overhead**.
>
> **GPU memory**.   Because in the second forward-backward pass we need to register the perturbation to the hidden embedding, Vaccine requires slightly more GPU memory usage.   We in the following show comparison results with the normal finetune (SFT) and a recent alignment-stage defense RepNoise [1] using different sizes of model.
>
> * Evaluation is done with an H100 with 80GB memory:
> |  | OPT-1.3b | OPT-2.7b | OPT-6.7b | OPT-13b |
> |---|---|---|---|---|
> | memory(SFT) | 5.586GB | 10.814GB | 25.469GB | 48.824GB |
> | memory (RepNoise) | 8.962GB | 17.453GB | 40.017GB | 76.027GB |
> | memory(Vaccine) | 5.596GB | 10.830GB | 25.492GB | 48.863 GB |
>
> Vaccine incurs slightly more GPU memory compared to SFT. For OPT-13B, only a marginal 0.039GB extra memory is induced compared to SFT.  In sharp contrast, RepNoise introduces 27.164GB extra memory overhead compared to SFT. With this result, it seems that Vaccine is superior in the resource-constrained scenario.
>
>
>
> **Training time**. Because Vaccine requires double forward-backward pass in each optimization step, the training time of Vaccine is approximately double compared to the SFT baseline. See the following table for our evaluation results.
>
> * Evaluation is done with an H100 with 80GB memory:
> |  | OPT-1.3b | OPT-2.7b | OPT-6.7b | OPT-13b |
> |---|---|---|---|:---:|
> | step time(SFT) | 0.06 | 0.08 | 0.09 | 0.12 |
> | step time(RepNoise) | 0.14 | 0.19 | 0.2 | 0.29 |
> | step time(Vaccine) | 0.11 | 0.15 | 0.17 | 0.24 |
>
> As shown, Vaccine uses approximately 2x training time and RepNoise uses approximately 2.3x-2.4x training time compared to SFT.  Vaccine is more computation-efficient compared to RepNoise.
>
> It is also possible to accelerate Vaccine. Vaccine requires double training time, because in every step we need to first search for the perturbation, and then apply the perturbation to the model to do another forward-backward pass. To address the reviewer's concern, we propose **accelerated Vaccine**. The idea is simple- we search for the perturbation **not for every step**, but to do it for **every $\tau$ steps**. See Algorithm 2 in our attached pdf for details.     The following table shows the evaluation results.
>
> | Methods | Harmful score | Finetune accuracy | Step time |
> |:---:|:---:|:---:|:---:|
> | SFT | 59.20 | 94.40 | 0.12902s (1x) |
> | Vaccine | 50.40 (+0) | 92.40 | 0.24852s (1.93x) |
> | Accelerated Vaccine($\tau=100$) | 51.00 (+0.60) | 95.20 | 0.13140s (1.02x) |
> | Accelerated Vaccine($\tau=1000$) | 52.00 (+1.60) | 94.20 | 0.12956s (1.004x) |
> | Accelerated Vaccine($\tau=10000$) | 53.20 (+2.80) | 94.80 | 0.12934s (1.002x)|
> | Accelerated Vaccine($\tau=20000$) | 58.80 (+8.40) | 94.40 | 0.12902s (1x) |
>
> Our results surprisingly show that by searching perturbation for every 100 steps, Acceleated Vaccine is still able to achieve decent defense (harmful score is increased by a marginal 0.60), but the step time is significantly shortened. We hope accelerated Vaccine can erase the reviewer's concern about resource overhead.
>
> [1] Rosati D, Wehner J, Williams K, et al. Representation noising effectively prevents harmful fine-tuning on LLMs[J]. arXiv preprint arXiv:2405.14577, 2024. https://arxiv.org/abs/2405.14577

---

> > ### Comment · Reviewer_Pifs · 2024-08-12
> >
> > Thank you for the clarification and additional experiments. The resource overhead of vaccine is smaller than I expected. The author should consider adding this content to the article and making it open source. I have decided to raise my score.

---

> ### Author Response · Authors · 2024-08-12
> **Thanks for recognizing our effort in rebuttal!**
>
> Thank you for recognizing our effort in rebuttal, and also for increasing rating to reflect the addressed concern. We will definitely include the new results to the paper, and also open-source the accelerated Vaccine method.

---

### Official Review · Reviewer_a2cG · 2024-07-30

**Soundness:** 3
**Presentation:** 3
**Contribution:** 3
**Rating:** 6
**Confidence:** 3

**Summary:**

This paper introduces a novel phenomenon called harmful embedding drift, which occurs when a few harmful data points uploaded by users cause misalignment in the fine-tuned LLM. To combat this, this paper proposes a technique called Vaccine, which uses perturbation-aware alignment to produce invariant hidden embeddings. This method aims to maintain the alignment of LLMs even when fine-tuning on potentially harmful user data. The empirical results demonstrate that Vaccine improves the robustness of LLMs against harmful prompts while preserving their reasoning abilities on benign prompts.

**Strengths:**

The paper focuses on an interesting and important topic, proposing a method that performs well across diverse evaluations.

**Weaknesses:**

1. The Vaccine method introduces additional computational overhead due to the need for two forward-backward passes for each step of model optimization. Further evaluation should be shared to show the trade-offs between robustness and efficiency.
2. While the paper demonstrates the efficacy of Vaccine on several datasets (SST2, AG-NEWS, GSM8K, and AlpacaEval), the scalability to larger datasets might require further investigation.
3. The effectiveness of the Vaccine method depends on the noise intensity (ρ), and choosing the optimal value may not be straightforward. Further guidelines on selecting this parameter would enhance the practical applicability of the method. (btw, what is the training data in Table 6)

**Questions:**

1. How does the computational overhead introduced by the Vaccine method compare with other alignment techniques in terms of real-time performance and resource consumption?
2. Will all the fine-tuning jailbreak attacks cause harmful embedding drift? How to define the effectiveness bound of the proposed harmful embedding drift phenomenon?

**Limitations:**

The authors have discussed the limitations and further impacts in the appendix.

---

> ### Author Rebuttal · Authors · 2024-08-04
>
> **W1+Q1: Reource ovehead**
>
> **GPU memory**.   Because in the second forward-backward pass we need to register the perturbation to the hidden embedding, Vaccine requires slightly more GPU memory usage.   We in the following show comparison results with the normal finetune (SFT) and a recent alignment-stage defense RepNoise [1] using different sizes of model.
>
> * Evaluation is done with an H100 with 80GB memory:
> |  | OPT-1.3b | OPT-2.7b | OPT-6.7b | OPT-13b |
> |---|---|---|---|---|
> | memory(SFT) | 5.586GB | 10.814GB | 25.469GB | 48.824GB |
> | memory (RepNoise) | 8.962GB | 17.453GB | 40.017GB | 76.027GB |
> | memory(Vaccine) | 5.596GB | 10.830GB | 25.492GB | 48.863 GB |
>
> Vaccine incurs slightly more GPU memory compared to SFT. For OPT-13B, only a marginal 0.039GB extra memory is induced compared to SFT.  In sharp contrast, RepNoise introduces 27.164GB extra memory overhead compared to SFT. By this result, it seems that Vaccine are superior in resource-constrained scenarios.
>
> **Training time**. Because Vaccine requires double forward-backward pass in each optimization step, the training time of Vaccine is approximately double compared to the SFT baseline. See the following table for our evaluation results.
>
> * Evaluation is done with an H100 with 80GB memory:
> |  | OPT-1.3b | OPT-2.7b | OPT-6.7b | OPT-13b |
> |---|---|---|---|:---:|
> | step time(SFT) | 0.06 | 0.08 | 0.09 | 0.12 |
> | step time(RepNoise) | 0.14 | 0.19 | 0.2 | 0.29 |
> | step time(Vaccine) | 0.11 | 0.15 | 0.17 | 0.24 |
>
> As shown, Vaccine uses approximately 2x training time and RepNoise uses approximately 2.3x-2.4x training time compared to SFT.  Vaccine is more computation-efficient compared to RepNoise.
>
> **Accelerated Vaccine**. It is also possible to accelerate Vaccine. Vaccine requires double training time because in every step we need to first search for the perturbation, and then apply the perturbation to the model to do another forward-backward pass. To address the reviewer's concern, we propose **accelerated Vaccine**. The idea is simple- we search for the perturbation **not for every step**, but to do it for **every $\tau$ steps**. See Algorithm 2 in our attached pdf for details.     The following table shows the evaluation results.
>
> | Methods | Harmful score | Finetune accuracy | Step time |
> |:---:|:---:|:---:|:---:|
> | SFT | 59.20 | 94.40 | 0.12902s (1x) |
> | Vaccine | 50.40 (+0) | 92.40 | 0.24852s (1.93x) |
> | Accelerated Vaccine($\tau=100$) | 51.00 (+0.60) | 95.20 | 0.13140s (1.02x) |
> | Accelerated Vaccine($\tau=1000$) | 52.00 (+1.60) | 94.20 | 0.12956s (1.004x) |
> | Accelerated Vaccine($\tau=10000$) | 53.20 (+2.80) | 94.80 | 0.12934s (1.002x)|
> | Accelerated Vaccine($\tau=20000$) | 58.80 (+8.40) | 94.40 | 0.12902s (1x) |
>
> Our results show that by searching perturbation for every 100 steps, Acceleated Vaccine is still able to achieve decent defense, but the step time is significantly shortened. We hope accelerated Vaccine can erase the reviewer's concern about resource overhead.
>
> **W2: Scalaibility to larger datasets**
>
> It is a good idea to test Vaccine in another larger-scale fine-tuning dataset. Here we follow the setting form [2] to finetune the model on a WikiText-2, and we show the model's tradeoff between perplexity (smaller the better) and harmful score.
>
> |  | Harmful score |  -->| --> |  -->|  -->| Perplexity | --> |-->  | --> |  -->|
> |---|:---:|:---:|:---:|:---:|:---:|:---:|:---:|:---:|:---:|:---:|
> |  | p=0 | p=0.01 | P=0.05 | p=0.1 | p=0.2 | p=0 | p=0.01 | P=0.05 | p=0.1 | p=0.2 |
> | SFT | 54.60  | 55.60  | 58.00  | 59.00  | 63.40  | 26.08  | 26.11  | 26.29  | 26.61  | 27.27  |
> | Repnoise | 54.20  | 53.80  | 55.80  | 57.60  | 59.20  | 35.32  | 36.11  | 37.16  | 36.64  | 38.68  |
> | Vaccine | 46.60  | 46.20  | 48.40  | 50.80  | 55.40  | 34.36  | 34.45  | 34.67  | 34.85  | 35.77  |
>
> As shown, Vaccine achieves smaller harmful score but maintain a lower perplexity compared to RepNoise.
>
> **W3: Selection of hyper-parameter**
>
> Indeed, Vaccine relies on the noise intensity $\rho$ for providing an effective defense.  In Table 6, we use **SST2** as a fine-tuning task (which is our default setup) to demonstrate how choosing different $p$ affects the harmful score and finetune accuracy.  Below we show the data for providing intuition of how to select $\rho$.
>
>
> | Methods | $\rho=0.01$ | $\rho=0.1$ | $\rho=1$ | $\rho=2$ | $\rho=5$ | $\rho=10$ |
> |:---:|:---:|:---:|:---:|:---:|:---:|:---:|
> | HS | 54.40 | 56.80 | 54.40 | 49.00 | 46.20 | 44.20 |
> | FA | 94.40 | 95.00 | 94.40 | 93.60 | 92.80 | 89.00 |
>
> As shown, with a larger $\rho$, the harmful score will decrease but also at the cost of degrading finetune accuracy. In this case, before deploying Vaccine, one should use a validation dataset to test the model's finetune performance and harmful score, and select the best tradeoff that is acceptable to the application.
>
> Here we also want to highlight that we only have one hyper-parameter to tune. In comparison, the recent method RepNoise needs two hyper-parameters, and requires an additional harmful dataset in their assumption. The simplicity of Vaccine should be merited.
>
>
> **Q2: Will all the fine-tuning attack cause harmful embedding drift?**
>
> Honestly, we don't have an definitive answer to this question. At least the original attack, i.e., mixing normal harmful data in the fine-tuning data, will cause harmful embedding drift. There may be more advanced attacks that can circumvent the embedding drift, which, however, is still under-explored.   In the attached pdf, we also use t-SNE to illustrate the harmful embedding drift phenomenon,  which you may be interested in. Feel free to check it out.
>
>
> [1] Rosati D, Wehner J, Williams K, et al. Representation noising effectively prevents harmful fine-tuning on LLMs[J]. arXiv preprint arXiv:2405.14577, 2024.
>
> [2]  Li Y, Yu Y, Liang C, et al. Loftq: Lora-fine-tuning-aware quantization for large language models[J]. arXiv preprint arXiv:2310.08659, 2023.

---

> ### Author Response · Authors · 2024-08-12
> **A brief summary of our rebuttal**
>
> Hi Reviewer a2cG,
>
> Thanks for the very informative review, and also for recognizing the harmful fine-tuning that we focus on is an interesting and important topic. We want to get back to you to see whether our rebuttal addresses your concerns.
>
> Here we respectfully summarize our efforts to address your main concerns.
>
> 1. To show the **trade-offs between robustness and efficiency**, we design an accelerated Vaccine method. It is shown in the table in the rebuttal that this method has a better tradeoff between robustness and efficiency -- harmful score is increased by a marginal 0.60 while the training time can be shortened by 50%, compared to the original Vaccine.
>
> 2.  To show that our method can be **generalized to larger datasets**, we show additional experiments on WikiText-2 with a larger number of tokens. The results indicate that Vaccine can reduce harmful scores while maintaining the model's perplexity.
>
> 3. We show further guidelines on **selecting  hyper-parameter $\rho$** by conducting experiments. The general trend is that with a larger $\rho$, the harmful score will decrease but at the cost of degrading finetune accuracy.
>
> 4. We compare another **alignment-stage solution RepNoise** in terms of GPU memory and training time. Our experimental results reflect that Vaccine is superior to RepNoise regarding both GPU memory and training time.
>
> We hope that these efforts can fully address your concerns, and we are more than happy to discuss them with you!

---

> > ### Author Response · Authors · 2024-08-14
> > **Warm reminder of author-reviewer discussion deadline**
> >
> > Hi Reviewer a2cG,
> >
> > We sincerely thank you for the insightful review comments. As the deadline for the author-reviewer discussion is approaching (less than 8 hours), could you please have a check on our rebuttal? It would be nice if the rating could be slightly adjusted if you found our rebuttal can address your concern.
> >
> > Per your initial review, it seems that your main concern lies in the **resource overhead** of Vaccine. In the rebuttal, we provide more information on the resource overhead of Vaccine on different size of models. Our results indicate that the method requires 2x training time and a marginal extra GPU memory overhead (0.039GB).
> >
> > To further shorten the training time, we propose **accelerated Vaccine**, whose idea is to search for the perturbation not for every step, but to do it for every $\tau$ step. In this way, we can significantly shorten the training time. Our results show that the accelerated Vaccine is approximately 2x faster than the original Vaccine (approximately the same training time with SFT), while still achieving decent defense (less than 0.6% loss of harmful score).
> >
> > We are more than happy to discuss this and other concerns with you!

---

> > > ### Comment · Reviewer_a2cG · 2024-08-14
> > >
> > > Apologies for the late reply. Since my fellow reviewers are largely in favor of acceptance, I initially felt there wasn't much to discuss. However, after reading your rebuttal, I have decided to raise my score. Thank you, nice work.

---

> > > > ### Author Response · Authors · 2024-08-14
> > > > **Thank you for the feedback!**
> > > >
> > > > Thank you for the encouragement and all the insightful questions in the initial review!
> > > >
> > > > Pariticularly, for your question "Will all jailbreak attacks cause harmful embedding drift?", we indeed do not have a definitive answer. It is possible that we can use this as an insight to develop adaptive attack to Vaccine, which is quite interesting to explore. Again, thanks for your review for our paper!

---

### Author Rebuttal · Authors · 2024-08-04

We sincerely thank all the reviewers and the AC for their efforts in reviewing our Vaccine paper!  Per the initial review, all the reviewers hold a positive view of our paper. The compliments are too many to be counted, e.g., "(Vaccine) focuses on an interesting problem and have diverse evaluations" (**Reveiwer a2cG**), "have comprehensive empirical evidence, well-explained problem and methodology" (**Reveiwer Pifs**),   "can be a plug-and-play method in different application scenarios" (**Reveiwer CLDA**), and "a work that may be interesting to the broad community" (**Reviewer M5RT**).

However, we do observe that there are two important weaknesses mentioned by the reviewers that our paper is lacking:

-----------

1) **Resource/Training time overhead**.
All four reviewers have concerns on the deployment of Vaccine because it needs to double the training time for safety alignment. This extra overhead is required because Vaccine needs to do two forward-backward passes for each optimization step.

We value this comment and we generally believe that this extra resource overhead may hinder the scalability of the algorithm. Fortunately, we can present new results to show that **it is actually not necessary** to do two forward-backward passes for **each optimization step**! We present the **accelerated Vaccine** in Algorithm 2 in the attachment.  The evaluation result of  accelerated Vaccine is given in Table 10 in the attachment.



-----------
2) **Lack of intuitive visualization of harmful embedding drift.**
Reviewer CLDA mentions that the harmful embedding drift phenomenon, though interesting, lacks an intuitive visualization. Reviewer a2cG seems also to be curious about the phenomenon.

In response to this concern, we follow Reviewer CLDA's suggestion to use t-SNE to visualize the embedding. The visualization is available in Figure 5 in the attachment. The results indicate that the embedding drift of Vaccine is less serious, explaining its success in maintaining the alignment knowledge and resisting harmful fine-tuning.

-----------

Other relevant concerns are also addressed by answer to each individual reviewer.

---

> ### Author Response · Authors · 2024-08-09
> **Typo correction**
>
> As pointed out by Reviewer CLDA, there is a typo in the Algorithm 2 in the rebuttal pdf. Line 4 in the algorithm should be "if $t$ mod $\tau$ == 0" instead of "if $t$ mod $\gamma$ == 0". In this algorithm, we basically do the perturbation every $\tau$ steps instead of per step, in order to accelerate vanilla Vaccine. We apologize for the confusion.

---

### Decision · Program_Chairs · 2024-09-25

**Decision:**

Accept (poster)

**Comment:**

**Summary of the Paper**
This paper introduces "Vaccine", a perturbation-aware alignment technique that mitigates the effects of embedding drift during user fine-tuning. The technique progressively adds crafted perturbations during the alignment phase, enabling embeddings to withstand harmful perturbations from user fine-tuning data. Their experiments demonstrate that the method reduces harmful scores while maintaining fine-tuning accuracy.

**Summary of Reviews**
- Reviewer a2cG (Score 6 - Weak Accept): This reviewer commended the paper's focus on an important topic of safety and the method's performance across diverse evaluations. They noted that the method introduces additional computational overhead, limiting its practical use. They also considered the scalability of the method to larger fine-tuning datasets. The authors responded that Vaccine uses slightly more GPU memory compared to SFT, while a comparative alignment-stage defense RepNoise introduces significantly more memory overhead.
- Reviewer Pifs (Score 6 - Weak Accept): The reviewer commended the comprehensive evaluations of Vaccine and commented on the increased computational and memory overhead, which limits its practical deployment in resource-constrained environments.
- Reviewer CLDA (Score 6 - Weak Accept): The reviewer commended the authors' presentation of the method and the generality of the approach. They commented on how the method doubles training time and whether more baselines can be covered to contrast this method with other defenses. They also have reservations about using a black box moderation model for harmful scores, one of the primary metrics for evaluating this method.  The authors responded that there is no alternative benchmarking method for evaluating harmfulness and that compared to using closed-source models like GPT-4, using an open-source moderation model like BeaverTails is superior.
- Reviewer M5RT (Score 6 - Weak Accept): The reviewer commended the validation of this method on widely-used LLMs and the potential of this topic to be interesting to a broad community. They also noted the computational overhead and the weak performance of the model when the harmful ratio is high.

**Assessment**

This paper presents an alignment stage defense to withstand harmful user data during fine-tuning. Their method demonstrates an improvement over existing methods when applied to a broad set of models and fine-tuning tasks. Reviewers noted the increase in computational overhead required to deploy this method. The authors responded that while Vaccine uses 2x the training time and more GPU memory compared to SFT, another recent alignment defense RepNoise uses more memory and requires longer training time.

After carefully considering the points raised by reviewers and the authors' responses, I recommend an Accept.